**PLOS** | BIOLOGY

METHODS AND RESOURCES

# Image-based analysis of living mammalian cells using label-free 3D refractive index maps reveals new organelle dynamics and dry mass flux

**Patrick A. Sandoz**[1☯], **Christopher Tremblay**[1,2☯], **F. Gisou van der Goot**[1]*, **Mathieu Frechin**[2]*

**1** Global Health Institute, Life Sciences Faculty, EPFL, Lausanne, Switzerland, **2** Nanolive SA, EPFL Innovation Park, Ecublens, Switzerland

☯ These authors contributed equally to this work.
* gisou.vandergoot@epfl.ch (GvdG); mathieu@nanolive.ch (MF)

**Data Availability Statement:** All relevant data are within the paper and its Supporting Information files.

## Abstract

Holo-tomographic microscopy (HTM) is a label-free microscopy method reporting the fine changes of a cell's refractive indices (RIs) in three dimensions at high spatial and temporal resolution. By combining HTM with epifluorescence, we demonstrate that mammalian cellular organelles such as lipid droplets (LDs) and mitochondria show specific RI 3D patterns. To go further, we developed a computer-vision strategy using FIJI, CellProfiler3 (CP3), and custom code that allows us to use the fine images obtained by HTM in quantitative approaches. We could observe the shape and dry mass dynamics of LDs, endocytic structures, and entire cells' division that have so far, to the best of our knowledge, been out of reach. We finally took advantage of the capacity of HTM to capture the motion of many organelles at the same time to report a multiorganelle spinning phenomenon and study its dynamic properties using pattern matching and homography analysis. This work demonstrates that HTM gives access to an uncharted field of biological dynamics and describes a unique set of simple computer-vision strategies that can be broadly used to quantify HTM images.

## Introduction

Because of the transparent nature of a cell, microscopy techniques either use fluorescent markers or transform optical properties of the sample into an observable contrast (for example, phase contrast, differential interference contrast [DIC]). Each of these techniques comes with limitations. Photobleaching, phototoxicity, and interference of markers or ectopically expressed engineered proteins are major concerns. Classical label-free imaging techniques, while less perturbing, provide images with low information content because of poor contrast and resolution. In this context, holo-tomographic microscopy (HTM) [1] is of great interest because it can provide label-free, high-content images using a very low-power light source that

**Funding:** This work was supported by a grant from the Swiss National Science Foundation (SNSF) and the Swiss SystemsX.ch initiative evaluated by the Swiss National Science Foundation (LipidX). MF was funded by Nanolive SA. PS, CT, and FGvdG were funded by LipidX and SNSF. The funders had no role in study design, data collection and analysis, decision to publish, or preparation of the manuscript.

**Competing interests:** I have read the journal's policy, and the authors of this manuscript have the following competing interests: Dr. Mathieu Fréchin is an employee of Nanolive SA.

**Abbreviations:** CARS, Coherent Anti-Stokes Raman Scattering; CMOS, complementary metal oxide semiconductor; CP3, CellProfiler 3; DIC, differential interference contrast; EPFL, École polytechnique fédérale de Lausanne; ER, endoplasmic reticulum; Fis1, fission protein 1; GFP, green fluorescent protein; HTM, holo-tomographic microscopy; LBPA, lysobisphosphatidic acid; LD, lipid droplet; MEF, mouse embryonic fibroblast; mESC, mouse embryonic stem cell; MO, Microscope Objective; NA, Numerical Aperture; NAGTI, N-acetylglucosaminyltransferase I; OA, oleic acid; PFA, paraformaldehyde; RANSAC, Random Sample Consensus; RI, refractive index; RPE1, human retinal pigment epithelial cell line; SIFT, scale-invariant features transform; SRS, Stimulated Raman Scattering; YFP, yellow fluorescent protein.

generates no phototoxicity. The HTM device used in this study is based on quantitative phase microscopy [2–6], in which the object's complex wave field is encoded into a hologram. A partially coherent light beam generated by a laser diode (520 nm) is split into two beams to create a Mach–Zehnder interferometer setup [7]; the first one, called the object beam, interacts with the sample before being collected by a 60× objective, while the second one is the reference beam. The two beams are later brought to interference, and the resulting hologram is recorded on a complementary metal oxide semiconductor (CMOS) camera [1–4]. Moreover, the device combines this classical holographic approach with rotational scanning of the specimen [5] using a rotating arm equipped with a mirror as described in Fig 1A. The synthesis of the rotational series of scattering spectra is achieved using complex field deconvolution [8–10] in order to reconstruct a full 3D refractive index (RI) tomogram of even live [11] samples. Importantly, dynamically adjustable mirrors allow the optics of the HTM device to self-adjust during an acquisition experiment in order to adapt to sample changes such as medium evaporation [12]. While RI distributions have begun to be used in life sciences studies [13–16] and specific RI signatures for cell structures [17,18] have been partially analyzed, HTM systems can suffer from coherent noise created by scattered light from the rotational scanning mechanism [19]. This coherent noise perturbs the quality of the generated holograms and impedes spatial resolution, which results in reduced RI sensitivity and perceived image quality [20]. Thanks to its rotational scanning mechanism made of a rotating mirror, the HTM setup used in this study overcomes this limitation and allows, to the best of our knowledge, a unique characterization of cellular and organelle details by RI in space and time. The next challenge was to go beyond a qualitative approach and to use computer-assisted image analysis in order to harness the full potential of HTM images. We developed computer-vision strategies using FIJI [21], CellProfiler3 (CP3) [22], and custom code in order to investigate the evolution of number, shape, and dry mass flow of lipid droplets (LDs) and endocytic structures as well as of full mouse embryonic stem cells (mESCs) over division. Thanks to the inherent multiplexing capacity of HTM, i.e., its capacity to capture multiple organelles and cellular structures all at once, we could finally observe a multiorganelle rotation within mammalian cells. In order to characterize this phenomenon and pave the road for more in-depth studies of its molecular origins and role for the cell, we developed a computer-vision strategy adapted to quantify the rotation of complex patterns using pattern-matching and homography [23,24].

## Results

### Identification of cellular organelles at high resolution using HTM

Our HTM device is equipped with simultaneous holotomography and epifluorescence imaging (Fig 1A); we could therefore compare fluorescence imaging of subcellular markers with RI maps to determine whether specific organelles have a recognizable RI signature that would allow label-free analysis of organelle biology in control conditions or upon perturbations. Firstly, HTM performances were compared to other label-free methods such as fixed HeLa cells imaged with brightfield microscopy, in which contrast is created by the attenuation of the transmitted light; phase contrast [25], in which contrast is created by the capacity of the sample to scatter light; differential interference contrast (DIC) [26], which relies on two rays of polarized light, a reference and a sampling ray, that interfere to create contrast; and finally, HTM (S1 Fig). The plasma membrane, and to a lesser extent LDs and nuclei, could be distinguished by the established label-free technologies but were perfectly observable in 2D and 3D images provided by HTM (Fig 1B). LDs were also visible by HTM in human bone osteosarcoma epithelial cells (U2OS cell line), fibroblasts, human retinal pigment epithelial cells (RPE1), mouse embryonic fibroblasts (MEFs), murine hepatoma cells (Hepa1.6 cell line), chronic

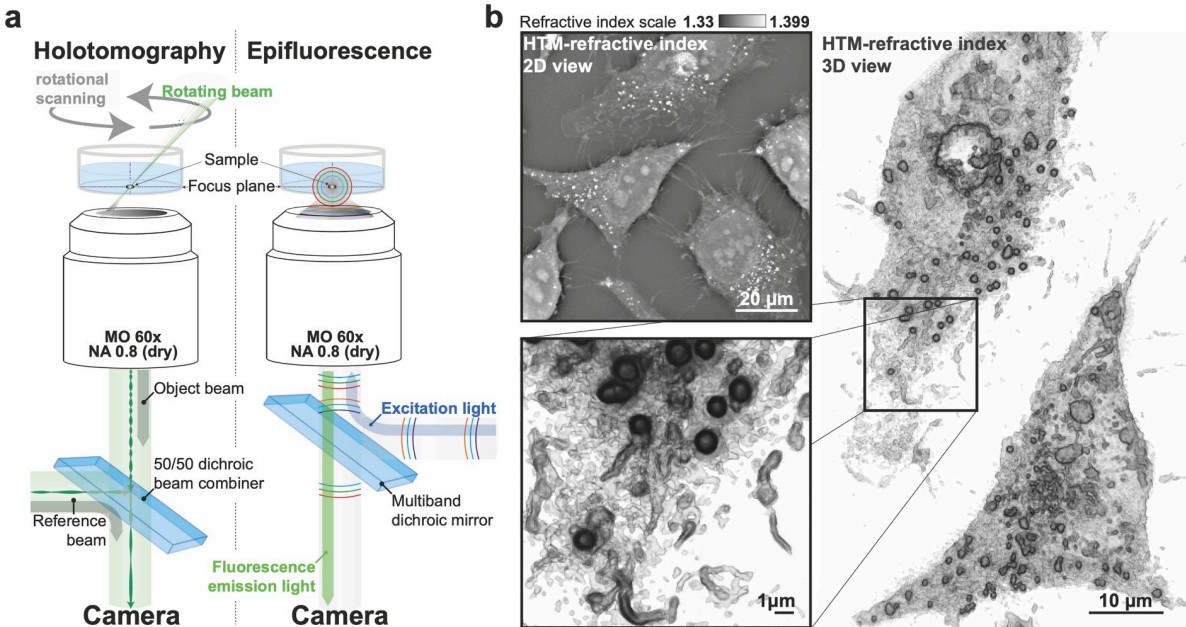

**Fig 1. HTM of subcellular structures.** (a) Scheme of the HTM setup coupled to epifluorescence used in this study. (b) 2D and 3D images of flat, unlabeled, and unperturbed HeLa cells taken with HTM demonstrate the necessity of a strategy to harness data. HTM, holotomographic microscopy; MO, Microscope Objective; NA, Numerical Aperture.

myelogenous leukemia cells (HAP1 cell line), and differentiated adipocytes (S2 Fig). In addition to LDs, HTM could clearly discriminate (in two and three dimensions) other intracellular details that could not be distinguished at all by the other methods (Fig 1B, S1 Fig), such as what we identify later as mitochondria. It is important to note that dehydrating fixation methods affect organelles' structural integrity, altering their RI and thus the image quality when observed with HTM. Paraformaldehyde (PFA), however, is a viable solution for observing fixed cells with HTM because it allowed the structural preservation of subcellular details (S3 Fig).

We next used the HTM/epifluorescence setup to compare the RI sections and the corresponding fluorescent signals of the cells under investigation. We first analyzed the largest organelle in the cell, namely the endoplasmic reticulum (ER) using KDEL-green fluorescent protein (GFP) as a fluorescent marker. Analysis of the corresponding RI map (S4A Fig) and the quantitative evaluation of RI and fluorescent signal correlations (S5A and S5B Fig) indicated that the ER does not lead to an RI signal in control cells. Similarly, using GFP-tagged N-acetylglucosaminyltransferase I (NAGTI-GFP) as a fluorescent marker, we found that the Golgi apparatus is not detected by HTM (S4B Fig). The fact that the ER and the Golgi are "silent" in terms of RI mapping is advantageous given that the ER is present throughout the cell and its detection would have hampered the proper analysis of any other organelle. In contrast, labeling of cells with the mitochondrial marker Mito-yellow fluorescent protein (YFP) (Fig 2A) or lipid marker Bodipy (Fig 2B) revealed that mitochondria and LDs provide a clearly distinguishable RI signal in two and three dimensions.

This visual conclusion was confirmed using Kolmogorov–Smirnov (S5A Fig) and Pearson coefficient tests (S5B Fig). It was also confirmed by the comparison of Mito-YFP and Bodipy fluorescent signals with independent human expert labeling of mitochondria and LDs in RI maps that shows over 90% of overlap (S5C Fig), altogether validating a specific correlation of the RI and fluorescent signals.

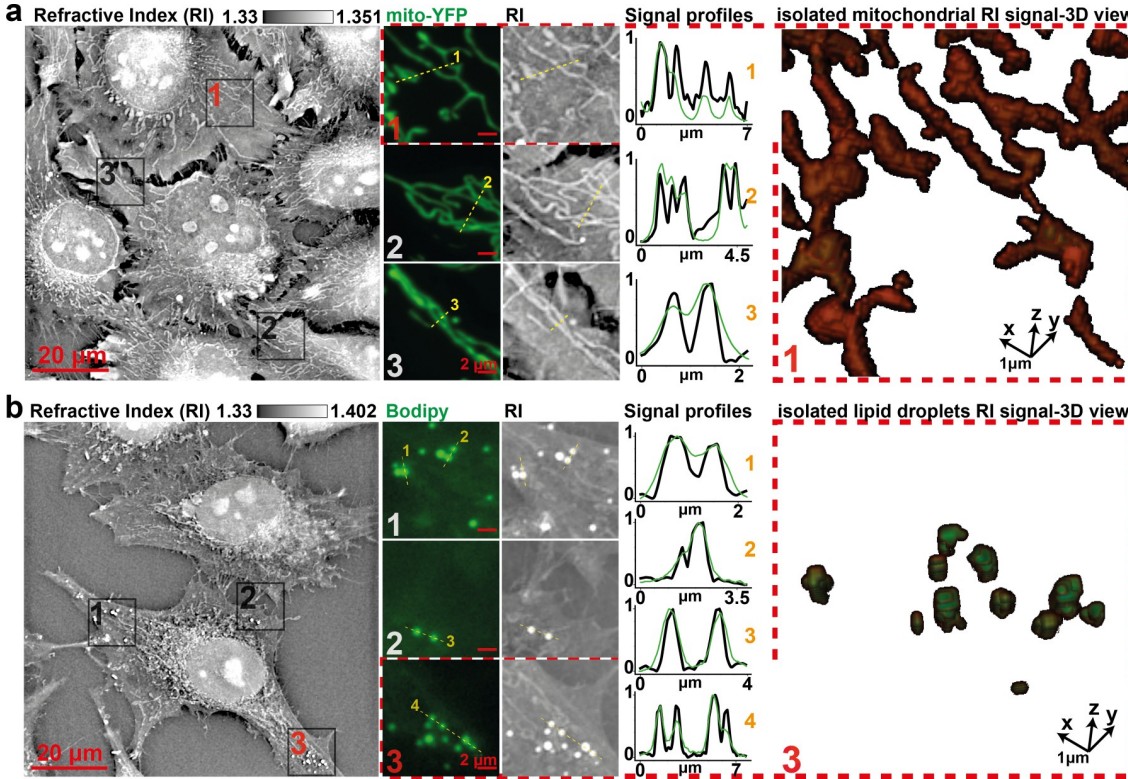

**Fig 2. Comparison of specific fluorescent signals and RI map.** Visual and profile comparison, as well as 3D view, of (a) HeLa cells' RI map to mitochondria-specific fluorescent signal (Mito-YFP) or (b) to an LD-specific fluorescent signal (Bodipy) shows specificity and gain in resolving power. Profiles were normalized between 0 and 1 for proper scaling. LD, lipid droplet; RI, refractive index; YFP, yellow fluorescent protein.

Interestingly, while in epifluorescence (or in a microscope using the same optics), such subcellular structures are being diffraction-limited to a maximum of resolution of $\eta_{fluo}^{Abbe} \approx 325nm$, our HTM's theoretical resolution limit [11] allows for $\eta_{HTM}^{Abbe} \approx 162nm$. Accordingly, the signal profiles in Fig 2A and 2B show an improvement in resolving subcellular structures using our RI measurement over the conventional epifluorescence signal. Thus, the resolution of mitochondrial tubules down to 186 nm by the used HTM setup (S6 Fig) demonstrates a subdiffraction imaging regime with respect to $\eta_{fluo}^{Abbe}$. To further confirm the ability of HTM to image mitochondria, we promoted mitochondrial fission by overexpressing the mitochondrial fission protein 1 (Fis1) [27]. While cells expressing a mitochondria-targeted DsRed protein (S7A Fig) exhibit regular elongated and thin mitochondria, cells transfected with Fis1 presented perinuclear collapsed and round-shaped structures (S7B Fig). We finally performed time-lapse acquisitions of mESCs at a frequency of 4 images · min⁻¹ in order to confirm our capacity to observe individual mitochondrial fission and fusion events (S8 Fig and S1 Movie) over time-lapse experiments. All together, these investigations demonstrate that HTM is suited for the study of cellular organelles with great spatial and temporal resolution.

## Image-based analysis of LD dry mass dynamics

So far, dynamics of LDs have been visualized using fluorescence microscopy and Coherent Anti-Stokes Raman Scattering (CARS and the variant Stimulated Raman Scattering [SRS]) [28,29]. On the one hand, fluorescence microscopy of LDs using specific Bodipy dyes [28]

provides contrast and relative specificity but is phototoxic. This forces a reduction of both acquisition length and frequency in order to decrease the perturbation of LD dynamics. On the other hand, CARS allows for working label-free; however, it is a complex and costly technology that has specific constraints regarding signal specificity and temporal resolution [29]. The consequence of those limitations is a lack of finely resolved dynamics of the first moments of LD appearance and growth. HTM can provide such data and could contribute to a better understanding of the still-elusive process of LD formation [30]. HeLa cells were incubated with oleic acid (OA) to trigger the growth of LDs [31] while imaged using HTM at a regime of 1 RI volume $\cdot$ min$^{-1}$ (S2 Movie). All volumes were then projected into 2D RI images using FIJI (Fig 3A), a requirement to be able to use CP3's object detection modules. CP3 was used for a systematic and precise segmentation of LD objects over time (Fig 3B and 3C). Segmented objects were then used to extract the mean RI of each detected LD, and we considered LDs as spheres in order to calculate their dry mass content based on a linear calibration model [32]. After 3 hours of incubation with OA, lipid droplets increased approximately 16-fold in number (Fig 3B and 3D, top left plot), and each single LD contained approximately 6 times more dry mass on average (Fig 3B and 3D, bottom left plot). Thanks to the extremely low light power used in our HTM setup, we could perform high-frequency acquisitions without perturbing organelle biology and could highlight new, to our knowledge, dynamic features of cellular lipid accumulation.

Firstly, the mean dry mass contained per LD decreases in the first 20 to 25 minutes after OA treatment with a mean dry mass flow per detectable lipid droplet starting around $-1 \times 10^{-4}$ pg $\cdot$ min$^{-1}$ $\cdot$ LD$^{-1}$ and reaching a 0 net flow around 25 minutes to later stay positive around $1 \times 10^{-4}$ pg $\cdot$ min$^{-1}$ $\cdot$ LD$^{-1}$ (Fig 3D, bottom plots). To give concreteness to such numbers, and if we consider that only OA had a net contribution to LD growth in our experiment, $1 \times 10^{-4}$ pg $\cdot$ min$^{-1}$ $\cdot$ LD$^{-1}$ represents, on average, $2.1 \times 10^{4}$ molecules of OA processed and stored in each LD every minute.

Importantly, during these first 25 minutes, we also see that the number of LDs increases steadily (Fig 3D, top left plot) and is multiplied approximately 3-fold. We also observed that in our experiments, new LDs do not arise from the fission of preexisting LDs, contrary to the phenomenon observed in yeast [33]. We therefore see that OA loading on HeLa cells has a surprising effect: LDs become smaller but more numerous, suggesting strongly that preexisting LDs unload material into a common pool to prepare and feed new lipid storage units before growing them. This is, to our knowledge, a new observation that would support the model of a fluid connectivity between existing and nascent LDs through the ER membrane [34–36].

We then observe that the number of LDs reaches a plateau around 100 minutes; however, the total dry mass content of all LDs increases until 210 minutes, when OA loading starts to become toxic [37]. This nearly linear accumulation hides bursts in the average dry mass flux per LD (Fig 3D, bottom right plot, after 50 minutes). If LDs accumulated lipids asynchronously, the resulting average flux curve would be smooth because the random variations at the single LD level at each time point would compensate each other; on the contrary, those bursts we observe in the average flux of dry mass per LD suggest a synchronized acceleration of lipid storage in at least a significant subset of LDs. Such synchronization of fluxes could be achieved by sharing a common pool and are an additional argument in favor of a fluid connection of LDs through the ER.

## Image-based analysis of dry mass accumulation in the late endocytic pathway

The fact that mitochondria, which are dense in membranes, and LDs are readily observable by HTM led us to speculate that other lipid-rich structures could provide a clear RI signal. We

therefore induced the formation of a lipid-rich organelle and did so by treating HeLa cells with U18666A, a charged amino derivative of cholesterol [38,39] that can trigger lipid and cholesterol accumulation in the late endocytic pathway and mimic the phenotype of Nieman–Pick

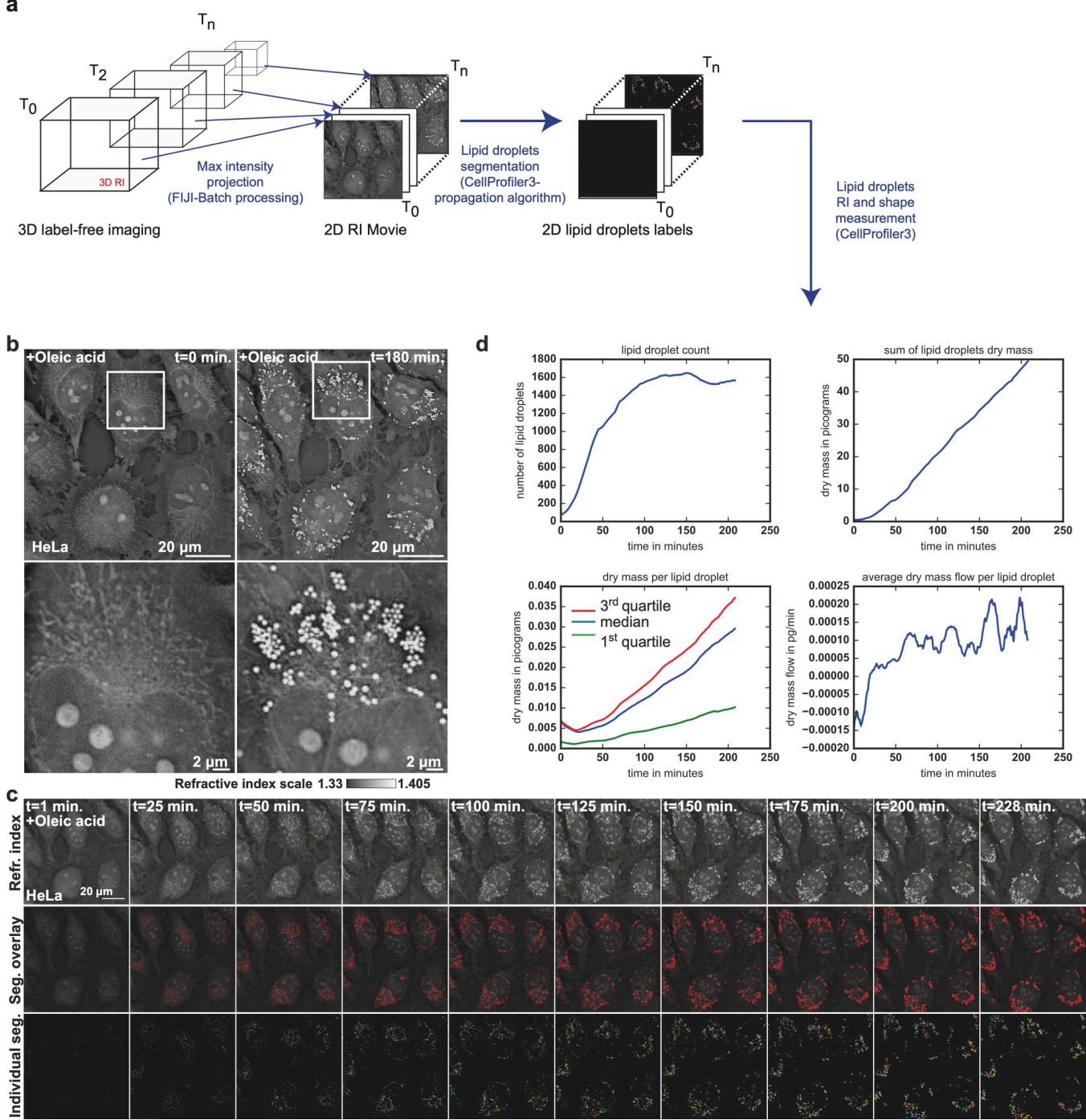

**Fig 3. Live HTM reveals LD dynamics.** (a) RI map processing and analysis strategy. (b) RI map of HeLa cells just before ($t = 0$ hours) or 3 hours after ($t = 3$ hours) loading with OA. (c) Time series of HeLa cells 2D RI maps, overlay of LD segmentations, and individual LD objects after treatment (OA) (related to S2 Movie). (d) Time traces of i) the increase of LD number in LD per minute, ii) the overall dry mass accumulation within LD or iii) per LD in picograms per minute, and iv) average dry mass flow observed per LD in picograms per minute[2]. HTM, holo-tomographic microscopy; LD, lipid droplet; OA, oleic acid; RI, refractive index.

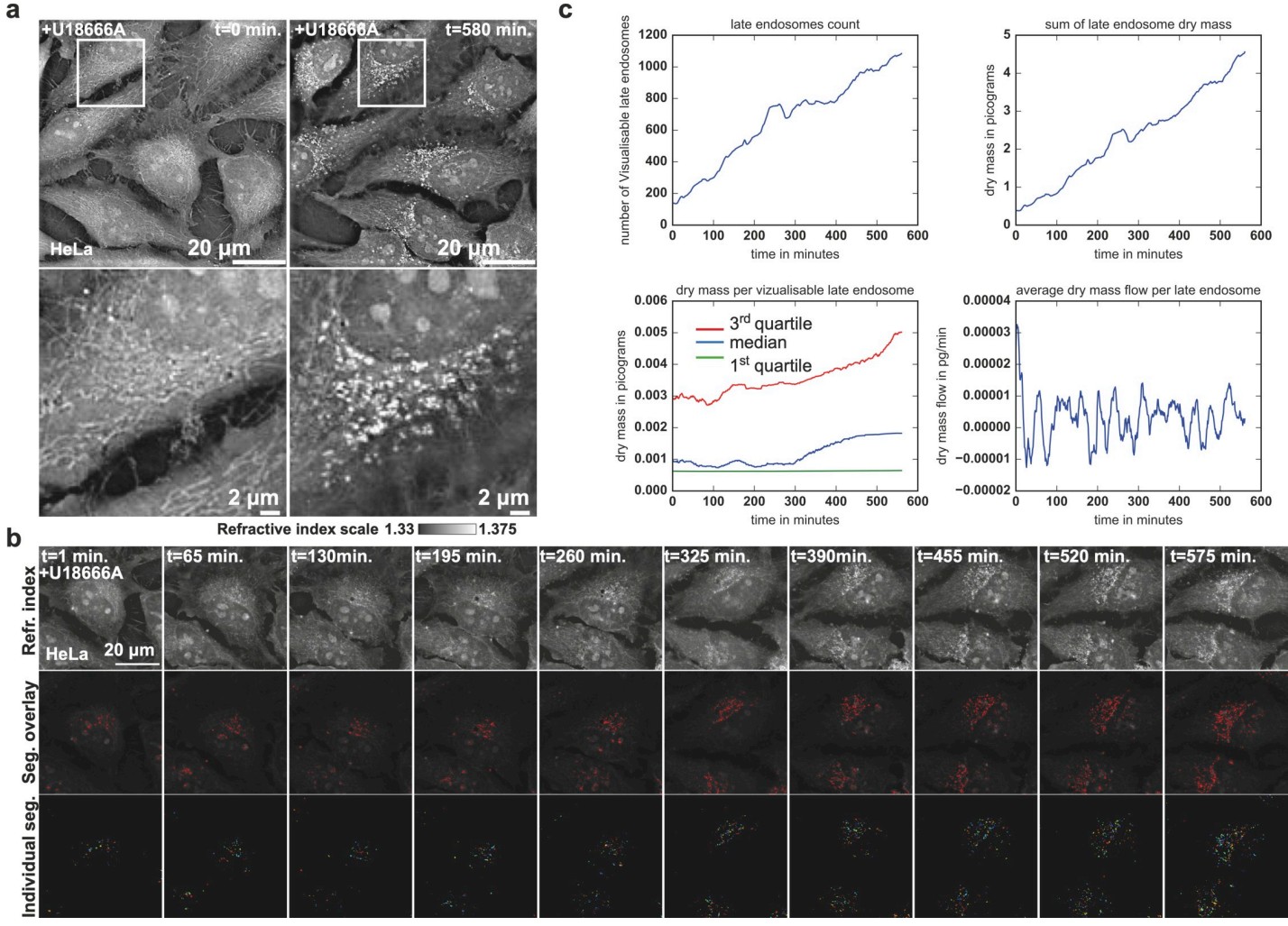

**Fig 4. Live HTM reveals perturbed trafficking dynamics.** (a) RI map of HeLa cells just before (*t* = 0 hours) or 10 hours after (*t* = 10 hours) treatment with the compound U18666A. (b) Time series of HeLa cells' 2D RI maps, overlay of LE segmentation, and individual LE objects after treatment (U18666A) (related to S3 Movie). (c) Time traces of i) the increase of LE number in LE per minute, ii) the overall dry mass accumulation within LE or iii) per LE in picograms per minute, and iv) average dry mass flow observed per LE in picograms per minute[2]. HTM, holo-tomographic microscopy; LE, late endosome; RI, refractive index; Seg., segment.

Type C disease [39]. We imaged HeLa cells for 10 h after treatment with U18666A using HTM at a frequency of 1 image · min⁻¹; we observed the appearance of perinuclear organelles with a well-defined RI signal (Fig 4A, S3 Movie). These structures are rich in cholesterol, as indicated by the filipin-positive staining, and colocalize with the late endosomal lysobisphosphatidic acid (LBPA) (S9 Fig). The accumulation of these U18666A-induced structures was followed and quantified using the image analysis procedure developed for LDs but adapted for smaller objects in CP3's detection modules (Fig 4B). The number of detectable lipid-rich endocytic structures increased approximately 6-fold over a 10-h treatment (Fig 4C, top left plot). This curve closely follows the total accumulation of dry mass in the structures (Fig 4C, top right plot); this is explained by the relative homogeneity in size of late endosomal structures over the whole experiment (Fig 4C, bottom left plot). Thanks to our high temporal resolution and computer-vision approach, we could observe details of dry mass fluxes inaccessible to the eye or to label-based time-lapse approaches. This analysis reveals that the accumulation of lipid and cholesterol in late endosomal structures is the result of oscillating fluxes of material that

successively accumulates and is removed. In accordance with current models of the effect of U18666A on lipid and cholesterol trafficking, our quantification suggests that U18666A does not strictly block recycling because negative fluxes suggest a recycling of a part of the accumulated material. It rather suggests that U18666A perturbs the equilibrium between recycling and uptake as suggested before [39], leading to progressive accumulation in late endosomal structures. Our approach should prove extremely useful for studying the contribution of given genes in the formation of aberrant late endosomal structures in lysosomal storage diseases or to find drugs able to fight lysosomal storage diseases such as Niemann–Pick type C disease.

## Image-based analysis of cellular dry mass dynamics before, during, and after mitosis

We then investigated the potential of HTM combined with fine cell segmentation to study the volume and dry mass of mESCs over mitosis. Such investigation is a technical challenge [40,41]. Recent studies have investigated mammalian cells' volume over mitosis using a wide range of techniques, from confocal microscopy to fluorescence exclusion method [42–45], but not with holotomography, which has the advantage of showing fine cellular details within high-quality RI maps directly in three dimensions. Some studies [44,45] show that mammalian cells' volume diminishes during mitosis, while others [42] demonstrate that various mammalian cell lines can swell during mitosis. Even if they lead to contradicting results, those studies show the importance of a proper object detection, and our approach focuses on improvements at that level thanks to HTM combined with state-of-the art object detection rather than concluding on mitotic volume dynamics, which would require a study on its own. We imaged mESCs at a frequency of 2 RIs volume · min$^{-1}$; all volumes were projected into 2D RI images using FIJI (Fig 5A) in order to be able to use CP3, much like the procedure developed for LDs. Cell objects were detected using the propagation algorithm of CP3. This 2D object segmentation on projected RI maps was then reapplied on each 3D volume, slice by slice, and adapted using the RI threshold defined during the 2D segmentation step (Fig 5B). This seeding procedure using a 2D premask ensures a continuity over the z dimension because CP3 is not yet ready for 3D object detection over large data sets.

We could extract the volume and RI map of each cell object contained in time-lapse experiments and perform object tracking with CP3 to be able to analyze the cell's features over time (Fig 6A, S4 Movie). The details of the segmentation and tracking procedure show that the image analysis process is able to catch even fine details of cellular protrusions, separate objects properly, and keep their identity with satisfying precision over time (Fig 6A and 6B). Quantifications show that the cells have a similar volume evolution until cytokinesis, and such trends are easily observable in S1 Movie. Their volume increases steadily before mitosis to reach approximately 6,000 μm$^3$; it then quickly reaches 1,500 μm$^3$ before daughter cells separate and spread out again. The mean dry mass, calculated from the RI based on a linear calibration model [32], increases during the rounding of the cell from approximately $25 \times 10^{-2}$ pg · μm$^{-3}$ to $6 \times 10^{-2}$ pg · μm$^{-3}$ before going back to $25 \times 10^{-2}$ pg · μm$^{-3}$. More surprisingly, the total dry mass of cells decreases during mitosis; because we are confident in the precision of our segmentation, this observation can be interpreted in two ways: i) a significant amount of cellular material is excluded from the cell during the mitotic roundup, in direct contradiction with recent data showing that cell mass accumulates also during mitosis [46], and this observation could be specific to the observed mESCs; or ii) the linear calibration model [32] linking RI and dry mass does not apply during mitosis because of dry mass reorganization [47] or even phase transition.

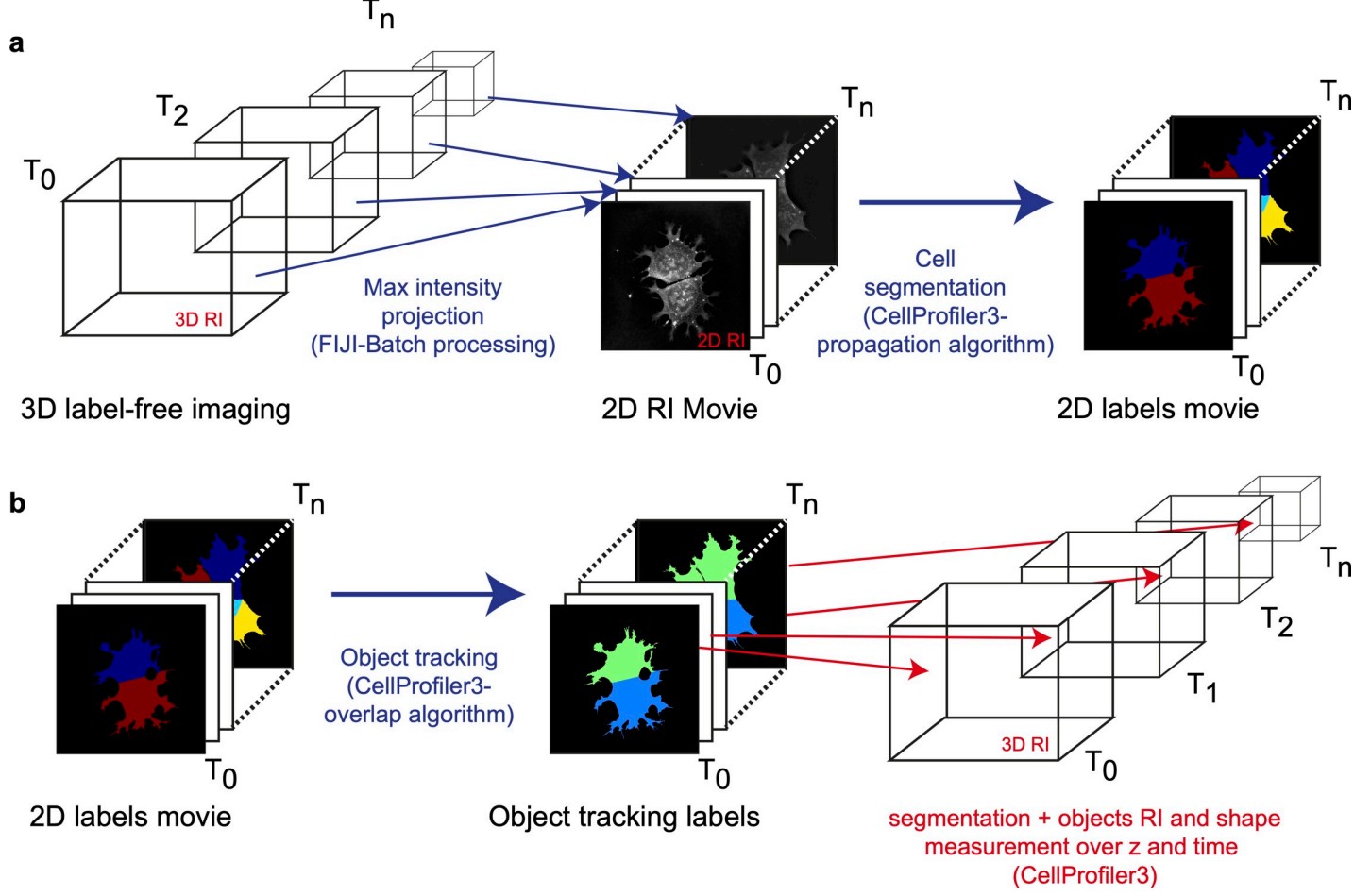

**Fig 5. Analysis of mESC dry mass dynamics over cell division: computational analysis workflow.** (a) 3D RI maps collected over an entire cell cycle are projected as 2D RI maps. Entire cells are detected in each frame using the propagation algorithm of CP3. (b) Cell tracking and measurement over z and time dimension in CP3. Masks are adapted at each z slice using the threshold defined during the cell detection in (a) (related to S4 Movie). CP3, CellProfiler3; mESC, mouse embryonic stem cell; RI, refractive index.

### Analysis of nuclear rotation using feature matching and homography

Our unique capacity to observe multiple cellular organelles all at once within mESCs and over long time periods allowed us to capture a phenomenon of organelle spinning that occurs before mitosis. Rotations around an axis perpendicular to the substratum was observed for the first time in the 1950s in epithelial cells, nerve cells [48], and HeLa cells [49] and was largely ignored until the mid-1980s, when the concept of nuclear rotation per se was challenged, with two proposed phenomena explaining the observations made at that point in time. The first model was based on a nucleoli displacement combined with karyoplasm streaming [50], in which only the intranuclear material would rotate and not the nuclear membrane; the second model supported a full nuclear rotation helped by the cytoskeleton [51,52] and with a major role of dynein in the process [53]. We observe here in mESCs that such a cellular event involves at least the entire nucleus, including the nuclear membrane and nucleoli that conserve their shape, as well as LDs, which supports the model of a rotation of the entire nucleus [51] and extends it with a concomitant cytosolic streaming revealed by LD rotation (Fig 7A, S5 Movie). To pave the road towards a better understanding of such rotational phenomena or of

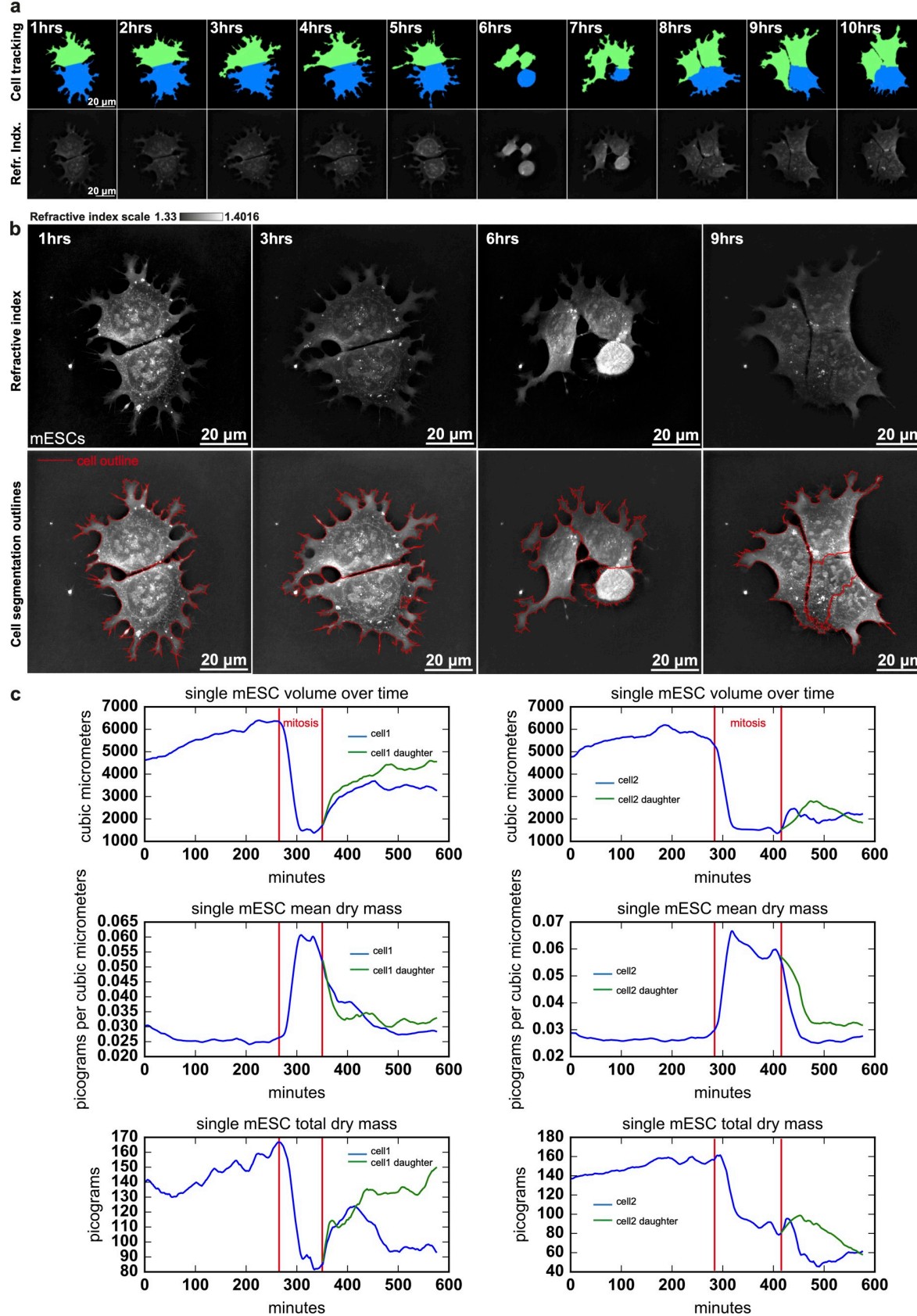

**Fig 6. mESC dry mass dynamics over cell division suggest significant dry mass loss during mitosis.** (a) Time series of mESC RI maps and tracking masks obtained with CP3. Green and blue signify the lineage. (b) Close-up view of 1, 3, 6, and 9 hours mESC RI maps and related segmentation overlaying details. (c) Time traces of (row 1) single mESC volume, (row 2) single mESC mean dry mass, and (row 3) single mESC total dry mass for (column 1) mESC 1 or (column 2) mESC 2 and their daughters (related to S1 and S4 Movie). CP3, CellProfiler3; mESC, mouse embryonic stem cell; RI, refractive index.

milder nuclear movement such as those seen in migration associated nuclear alignments [54,55], we developed a computer-vision approach in Python to help quantify the characteristics of the observed rotations (Fig 7B). This method is based on the use of scale-invariant features transform (SIFT) descriptors [56] originally developed for matching different views of an object or of a scene. The SIFT feature matching was done between the 2D RI maps at T and T + 1 from a time-lapse experiment displaying cells with rotating organelles (Fig 7C). Those matching features were then used to define a fitting homography matrix describing the transformation of T into T + 1 (Fig 7D). To do so, we used the Random Sample Consensus (RANSAC) method [57–59]. We finally extracted the rotation angle from the homography matrices and represented them in linear and rose plots in order to observe the rotational features of the phenomenon (Fig 7E and 7F). Our quantifications suggest that the phenomenon can vary greatly in time, from 10 hours to 45 minutes, but will only be stopped by mitosis. It varies also greatly in rotation magnitude, from 80 to 700 degrees, demonstrating that our approach can catch subtle nuclear movements as slow at 0.2 degree · min$^{-1}$ or relatively quick rotations as fast as 13 degrees · min$^{-1}$ (S10 Fig). It is common to observe pauses and bursts in rotations, with a homogeneity in rotation speed. In fact, our data set does not display oscillations in the rotational behavior; we observed only one direction reversal on the most limited rotation we recorded. Nuclear rotations have recently been shown to be involved in the dormant state of mouse oocytes [60] and to be key in the migration of fibroblasts [54]; in this context, we expect our approach to be a useful tool for characterizing the molecular and cellular details of this organellar rotation with, at its center, a full nuclear rotation.

## Discussion

In this work, we demonstrate that state-of-the-art HTM is suitable for the study of cellular organelles. Its label-free nature, combined with its excellent resolution and contrast and low phototoxicity, allows for imaging of organelles that have lipid membranes such as LDs, mitochondria, and pathologically lipid-loaded structures over long periods of time with great details and without sacrificing high temporal resolution. Importantly, HTM has the inherent capacity to capture multiple biological objects and phenomena simultaneously, a property also called multiplexing. This peculiar feature allows for observing complex cellular dynamics such as organellar rotations that implicate many subcellular structures. The true limitation of HTM lies, however, in its main strength: because it works label-free, the produced images are extremely rich and crowded with information that the researcher must sort out in an exploitable way. To this extent, we developed a set of image analysis approaches that are adapted for typical HTM label-free images and are easily reusable because they rely on existing and free image analysis software as well as on simple Python code. With such resources, we want to demonstrate that label-free holotomography is ready for integrating quantitative biology research strategies and that it can provide new insights in a broad range of biological dynamics. Our quantitative study of LDs identified insightful dynamics for understanding the process of early LD formation: we observe that newly forming LDs, all budding at the ER [30], are created at the expense of older LDs' material; this observation implies that LDs must exchange material over relatively short timescales, and the ER offers the perfect media for that purpose [36]. We also observe bursts in dry mass fluxes within LDs, suggesting a

**a**

**b**

Feature matching + homography

**c**

Linear plot of rotation over time

**d**

$$= \begin{bmatrix} h_{00} & h_{01} & h_{02} \\ h_{10} & h_{11} & h_{12} \\ h_{20} & h_{21} & h_{22} \end{bmatrix} \times$$

Rose plot of rotation over time

**e**

**f**

**Fig 7. Quantification of premitotic intracellular rotation using feature matching and homography.** (a) Time series of mESC RI maps showing nuclear rotation (red arrow) and LD rotation (yellow arrow and circles) before mitosis. (b) Computational workflow for RI image processing and quantification of rotation over time. (c, d) Homography matrices match resembling images subjected to translations and rotations. (e, f) Plot and rose plot of intracellular rotations over time show the direction, extent, accelerations, and pauses of the observed rotation (related to S5 Movie). LD, lipid droplet; mESC, mouse embryonic stem cell; RI, refractive index.

synchronization of fluxes in at least a subset of them, which could also be greatly facilitated by a continuity in their membrane through the ER. These observations are in support of a fluid and continuous connectivity between LDs [34–36]. Perturbing the link between LDs and the ER [61] while doing similar time-lapse experiments and quantifications of LD growth would provide new details on the connectivity existing between LDs and nascent ones or between growing LDs. More generally, the combination of our approach with genetic and chemical perturbations of the various machineries [30,36] identified for controlling LD maintenance and biogenesis could greatly improve our understanding of LDs.

We applied a similar approach to record fine dynamic features of the late endosomal accumulation of lipid and cholesterol. This phenomenon was triggered with the compound U18666A, which is classically used to emulate the Nieman–Pick type C phenotype [39]. Our investigation of dry mass fluxes in particular extends our knowledge of the effect of U18666A. We observe that this compound triggers an accumulation of material in late endosomal structures by shifting the equilibrium between recycling and uptake. We observe that recycling still occurs because the average dry mass flux per object oscillates over time and regularly takes a negative value, but not to such an extent that it could counterbalance accumulation. We demonstrate that our simple label-free approach combining HTM with proper image analysis gives access to unique features such as vesicle shape, size, number, and accumulation speed and dry mass amounts and fluxes. Importantly, this can be performed with minimal sample preparation, removing labeling- and phototoxicity-induced perturbations to which lipidic structures are very sensitive [62,63]. We believe that our work addresses a need for relevant, quantitative readouts that could accelerate the discovery of genes implicated in lysosomal storage diseases and the identification of drugs able to fight them [64].

We also developed a set of image analysis procedures that allow us to segment and follow entire cells and applied them to track mESCs before, during, and after mitosis. This task is a technical challenge [40,41] that requires specific devices and methods [42–45], which led to contradicting results regarding what happens to cell size during mitosis [42,44,45]. We demonstrate here that HTM images provide a sufficient level of details for performing very fine cellular segmentation and tracking using CP3, which opens up new possibilities for studying an entire cell's dynamics.

We finally observed striking organellar rotations within mESCs as well as in preadipocytes. Quantifying them was a challenge because, to the best of our knowledge, no preexisting tool in image analysis software would propose an integrated way to measure a coordinated multiobject rotation. We adapted a computer-vision strategy developed for the matching of different views of macro-objects [56] in order to detect and match rotating objects [57–59] in HTM time-lapse experiments. Such an approach has the advantage of working without prior object segmentation and is therefore well-adapted to the treatment of HTM images, which show complex textures composed of multiple biological objects. This simple strategy allowed us to contribute significantly to the understanding of the phenomenon of organellar rotation. Firstly, we could conclude that the entire nucleus rotates and not only the karyoplasm [50]. Furthermore, we could show that such rotation extends to the cellular space because we also observe a rotation of LDs, which could possibly be the consequence of a broader cytoplasmic streaming. Secondly, we characterized dynamic features of unperturbed rotations; observing

those features such as speed, acceleration, rotation length, and time with specific perturbations [51–53] will allow a better understanding of the molecular origin of the phenomenon and of its role for the cell's physiology, particularly in fundamental functions such as the dormant state of germ cells [60].

## Methods

### Cell cultures and seeding

Cells were cultured in MEM complemented with 10% FBS, 1% Pen/Strep, 1% L-glutamine, and 1% nonessential amino acids. Cells were seeded for 24 h at low concentration on glass bottom FluoroDishes of 25 mm and 0.17 mm thickness (World Precision Instruments Inc., Sarasota, FL, USA). For mESCs, FluoroDishes were first coated with Vitronectin following the manufacturer's protocol.

### Transfection

Cells were retrieved with trypsin from tissue culture dishes and seeded in FluoroDishes. After 24 h, the medium was changed, and the cells were transfected using Fugene (Promega, Madison, WI, USA) according to the manufacturer's protocol.

### Cell fixation

Fixation was performed with PFA or cold methanol. For PFA fixation, cells were first washed 3× with PBS. Then, 2 mL of PFA (3%) was added for 30 minutes at 37˚C. Afterwards, dishes were washed 3× with PBS. Quenching of the preparations was performed with 50 mM $NH_4Cl$ in PBS at room temperature for 10 minutes before 3× PBS washes. Permeabilization was performed using 0.1% Triton X100 for 5 minutes at room temperature. In the case of methanol fixation, cells were washed 3× with PBS before adding precooled methanol at -20˚C for 4 minutes. Cells were washed 3× with PBS after fixation.

### Immunofluorescence

Cells were seeded in FluoroDishes for 48 h (if transfected, including transfection) prior to fixation and permeabilization. Overnight blocking was performed in PBS with 0.5% BSA. Primary and secondary antibodies were applied for 30 minutes at room temperature each with in between 3× washes of PBS-0.5% BSA for 5 min. Finally, the preparation was washed again 3 times with PBS-0.5% BSA and postfixed for 15 min at room temperature with 3% PFA, followed by 3× PBS washes. Hoechst (Invitrogen, Carlsbad, CA, USA) was used at 2 μg/mL for 30 minutes at room temperature, Bodipy (Invitrogen) at 1 μg/mL for 30 minutes in physiological conditions. Filipin (Sigma-Aldrich, St. Louis, MO, USA) was used at a dilution of 1:50 (from stock 50 μg/ml).

### Drug treatments

U18666A was applied to the cells at a dilution of 1:2,000 (from a stock of 10 mg/ml). OA was applied at a dilution of 1:5,000 (from stock 1 mg/mL).

### Imaging

HTM, in combination with epifluorescence, was performed on the 3D Cell-Explorer Fluo (Nanolive, Ecublens, Switzerland) using a 60× air objective (NA = 0.8) at a wavelength of λ = 520 nm (Class 1 low power laser, sample exposure 0.2 mW/mm$^2$) and USB 3.0 CMOS Sony

IMX174 sensor, with quantum efficiency (typical) 70% (at 545 nm), dark noise (typical) 6.6 e⁻, dynamic range (typical) 73.7 dB, field of view $90 \times 90 \times 30$ μm, axial resolution 400 nm, and maximum temporal resolution 0.5 3D RI volume per second. The theoretical sensitivity is $2.71 \times 10^{-4}$. The correlative acquisitions with brightfield, phase contrast, and DIC were done on an Axiovert 200M (Zeiss, Oberkochen, Germany) using a 63× objective (NA 1.4).

### Live cell imaging

Physiological conditions for live cell imaging were reached with a top-stage incubator (Okolab, Pozzuoli, Italy). A constant temperature of 37˚C and an air humidity saturation as well as a level of 5% $CO_2$ were achieved throughout the acquisitions.

### Image analysis of LD dynamics

An export was performed within the software STEVE, which controls the HTM microscope, to transform RI volumes into .tiff format. By doing so, RI volumes can be read by the software FIJI. The exported 3D tiffs must be in float format to keep the explicit RI for each voxel value. The 3D RI volumes in .tiff format were then processed in batch within FIJI for performance purposes. 3D RI volumes were transformed into 2D RI maps using maximum intensity projections and were also saved as .tiff files. The resulting series of 2D frames could then be processed using CP3, which does not support full 3D data analysis yet. The CP3 pipeline was designed to load each 2D RI map, segment the contained objects using the primary objects detection module, and extract area, shape, and intensity features using the measurements modules. A critical point for proper object detection was to use a manual threshold value. While automatic threshold detection is suited for most fluorescent microscopy images, whose signal dynamics are related to a plethora of factors, RI images are quantitative and depend only on the nature of the biological object that is observed. Therefore, the threshold can be entered as a fixed value, in our case 1.354, to ensure relative stability of this first step of the detection procedure. The object size limits that we entered were designed to encompass the full spectrum of potential LD diameters from 1 to 7 pixels. The segmented objects were finally used to extract the area and the mean RI value of each LD in each frame of the time-lapse experiment. The data were exported as a .csv file into a Python environment, in which we used the extracted diameter of each LD to calculate its spherical volume; we could then calculate each LD dry mass content using a well-established linear calibration model [32]. The data were finally plotted with the Python library matplotlib.

### Image analysis of late endosomal accumulation of cholesterol and lipids

An export was performed within the software STEVE, which controls the HTM microscope, to transform RI volumes into .tiff format. By doing so, RI volumes can be read by the software FIJI. The exported 3D tiffs must be in float format to keep the explicit RI for each voxel value. The 3D RI volumes in .tiff format were then processed in batch within FIJI for performance purposes. 3D RI volumes were transformed into 2D RI maps using maximum intensity projections and were saved also as .tiff files. The resulting series of 2D frames could then be processed using CP3, which does not support full 3D data analysis yet. The CellProfiler3 pipeline was designed to load each 2D RI map, segment the contained objects using the primary objects detection module, and extract area, shape, and intensity features using the measurements modules. A critical point for proper object detection was to use a manual threshold value. While automatic threshold detection is suited for most fluorescent microscopy images, whose signal dynamics are related to a plethora of factors, RI images are quantitative and depend only on the nature of the biological object that is observed. Therefore, the threshold can be

entered as a fixed value, in our case 1.347, to ensure relative stability of this first step of the detection procedure. The object size limits were small and narrow from 1 to 3 pixels. The empirical process of finding the right values is fundamental to avoid detection of large clumps or the exclusion of unusually big structures. The segmented objects were finally used to extract the area and the mean RI value of each late endosomal structures in each frame of the time-lapse experiment. The data were exported as a .csv file into a Python environment, in which we used the extracted diameter to calculate its spherical volume; we could then calculate the dry mass content using a well-established linear calibration model [32]. The data were finally plotted with the Python library matplotlib.

### Image analysis of mESCs

An export was performed within the software STEVE, which controls the HTM microscope, to transform RI volumes into .tiff format. By doing so, RI volumes can be read by the software FIJI. The exported 3D tiffs must be in float format to keep the explicit RI for each voxel value. The 3D RI volumes in .tiff format were then processed in batch within FIJI for performance purposes. 3D RI volumes were transformed into 2D RI maps using maximum intensity projections and were saved also as .tiff files. The resulting series of 2D frames could then be processed using CP3, which does not support full 3D data analysis yet. The CP3 pipeline was designed to load each 2D RI map and to rescale them such that intracellular details do not perturb the global cellular detection; to do so, a simple rescaling from 0 to 1 of RI values between 1.32 and 1.34 was required. The subsequent segmentations of the cell objects were done using the secondary objects detection module based on primary objects defined a priori using the manual Primary Object detection module. The area, shape, and intensity features were then extracted using the related measurements modules on the unmodified RI maps. We preferred using a manual threshold value within the secondary object detection module, together with the propagation algorithm, for the best cellular segmentation. While automatic threshold detection is suited for most fluorescent microscopy images, whose signal absolute values are related to a plethora of factors, RI images are quantitative and depend only on the nature of the biological object that is observed. Therefore, the threshold can be entered as a fixed value because images were rescaled between 0 and 1; the manual threshold value we used is 0.89. The object size limits were between 80 and 250 pixels. The empirical process of finding the right values is fundamental to avoid the detection of under- or oversegmented cells. The objects obtained from the 2D RI frames segmentation were then used in CP3 to measure the objects in each z-frame at each time point, excluding any voxel whose RI was out of the RI range of the objects' RI defined at the previous step. The data were exported as a .csv file into a Python environment; the extracted area and RI values from each z-frame and time points were then reassembled to obtain 3D measurements, from which we could, for example, calculate total and mean dry mass using a well-established linear calibration model [32]. The data were finally plotted with the Python library matplotlib.

### Analysis of organellar rotations

An export was performed within the software STEVE, which controls the HTM microscope, to transform RI volumes into .tiff format. By doing so, RI volumes can be read by the software FIJI. The exported 3D tiffs must be in float format to keep the explicit RI for each voxel value. The 3D RI volumes in .tiff format were then processed in batch within FIJI for performance purposes. 3D RI volumes were transformed into 2D RI maps using maximum intensity projections and were also saved as .tiff files. The .tiff files opened as a stack in FIJI are then cropped to keep only one cell presenting a rotating nucleus and organelles in the field of view. The

resulting smaller 2D RI frames were then processed with a custom Python algorithm implementing the feature matching and homography procedures of the OpenCV library. The first step consists in detecting at least 10 SIFT descriptors [56] for each of the 2D RI frame-adjacent pairs based on Lowe's ratio test. The identical features from T to T + 1 were then used to define a fitting homography matrix describing the transformation of T into T + 1. To do so, we used the RANSAC method [57–59]. The rotation angle was then extracted from the homography matrices and plotted as regular and rose plots using the matplotlib Python library. For the details and reuse of the procedure, please see our adaptable Python algorithm (S1 Text).

## Supporting information

**S1 Fig. Comparison of HTM and other label-free microscopy techniques.** Brightfield, DIC, and phase contrast images show limited resolution and major artifacts. On the contrary, HTM images show low background and resolution high enough to observe details such as filopodia and mitochondria. DIC, differential interference contrast; HTM, holo-tomographic microscopy.
(TIFF)

**S2 Fig. Aspect of various cell lines when observed with HTM.** A wide range of cell lines are perfectly observable with holotomography. HTM, holo-tomographic microscopy.
(TIFF)

**S3 Fig. Impact of methanol or PFA fixation of HeLa cells on HTM results.** Methanol fixation affect organelles' structural integrity; PFA is a viable solution for observing fixed cells with HTM. HTM, holo-tomographic microscopy; PFA, paraformaldehyde.
(TIFF)

**S4 Fig. Comparison of specific fluorescent signals and RI map.** Visual comparison of HeLa cells' RI map to (a) Golgi apparatus fluorescent signal (NAGTI-GFP), (b) an ER fluorescent signal (KDEL-GFP), or (c) a late endosome accumulation signal (Filipin). ER, endoplasmic reticulum; GFP, green fluorescent protein; KDEL, XXX; NAGTI, N-acetylglucosaminyltransferase I; RI, refractive index.
(TIFF)

**S5 Fig. Correlation of various fluorescent and RI signal using various metrics.** (a) Bootstrapped Kolmogorov–Smirnov test of the distribution of RI values under specific fluorescent signals against the global cellular RI distribution. Statistical test $p$-values are plotted as a function of the stringency of the threshold used to define the fluorescence mask from a specific or random fluorescent object signal. (b) Pearson correlation of various fluorescent and RI signals. (c) Comparison of Mito-YFP and Bodipy fluorescent signals with independent human expert labeling of mitochondria and LDs in RI maps. LD, lipid droplet; RI, refractive index; YFP, yellow fluorescent protein.
(TIFF)

**S6 Fig. Full-width half-maximum quantification of minimal mitochondrial thickness using FIJI software.** The mitochondrial thickness has been measured within fibroblasts as the width observed at the half maximum of the mitochondrial signal distribution. The signal distributions along transversal lines have been defined within the FIJI software on 0 padded image and are represented in the red enlargement squares (red lines 1–3).
(TIFF)

**S7 Fig. Comparison of RI maps of cells when transfected with fluorescently tagged mitochondrial protein.** RI map of HeLa cells after transfection with (a) a neutral mitochondrial

DsRed marker or (b) with the mitochondrial fusion protein Fis1-GFP. Ds, XXX; Fis1, fission protein 1; GFP, green fluorescent protein; RI, refractive index.
(TIFF)

**S8 Fig. Live imaging of mitochondrial fission and fusion events in mESCs using HTM.** RI map of mESC times series showing mitochondrial fusion (blue arrow) and fission (yellow arrow) events. (related to S1 Movie). HTM, holo-tomographic microscopy; mESC, mouse embryonic stem cell; RI, refractive index.
(TIFF)

**S9 Fig. Comparison between RI map and late endosome or cholesterol fluorescent signal.** Compared to control, cells treated with U18666A show different perinuclear structures in the RI map that overlap with LBPA immunostaining and cholesterol filipin staining, indicating that the accumulation of cholesterol-rich late endosomes after treatment is observable in the RI map. LBPA, XXX; RI, refractive index.
(TIFF)

**S10 Fig. Quantification of premitotic intracellular rotation using feature matching and homography.** Plot and rose plot of intracellular rotation over time show the direction, extent, accelerations, and pauses of the observed rotation (related to S5 Movie).
(TIFF)

**S1 Movie. Live imaging of mESCs using HTM.** Movie length, 10 h; acquisition frequency, 4 images · min$^{-1}$. Related to Fig 6 and Extended Data Fig 6. This movie displays fine mitochondrial fusion and fission events as well as two full mitosis cycles. HTM, holo-tomographic microscopy; mESC, mouse embryonic stem cell.
(MP4)

**S2 Movie. Live imaging of LD growth in HeLa cells after OA loading using HTM.** Movie length, 3 h; acquisition frequency, 1 image · min$^{-1}$. Related to Fig 3. This movie displays LD growth that is quantified in the manuscript. HTM, holo-tomographic microscopy; LD, lipid droplet; OA, oleic acid.
(AVI)

**S3 Movie. Live imaging of late endosome accumulation in HeLa cells after U18666A treatment using HTM.** Movie length, 10 h; acquisition frequency, 1 image · min$^{-1}$. Related to Fig 4. This movie displays the lipid accumulations in late endosomes that are quantified in the manuscript. HTM, holo-tomographic microscopy.
(MP4)

**S4 Movie. Tracking segmentation labels of mESCs in S1 Video.** Movie length, 10 h; acquisition frequency, 4 images · min$^{-1}$. Related to Fig 6. mESC, mouse embryonic stem cell.
(AVI)

**S5 Movie. Live imaging of intracellular rotations in mESCs and mouse preadipocyte using HTM.** Movie length is indicated in each submovie square; acquisition frequency, 1 image · min$^{-1}$. Related to Fig 7 and Extended Data S10 Fig. HTM, holo-tomographic microscopy; mESC, mouse embryonic stem cell.
(MP4)

**S1 Text. Adaptable Python algorithm for feature matching and homography analysis.**
(TXT)

## Acknowledgments

We thank José Artacho from École polytechnique fédérale de Lausanne (EPFL) Imaging Platform for his great help with the phase contrast and DIC correlative acquisitions. We thank Sebastien Equis for sharing Fig 1A image material and helpful discussions and Lisa Polaro and Yann Cotte for helpful discussions. In addition, we thank Hubert Becker for hand-labeling of mitochondria and LDs, Aleksandra Mandic for providing mESCs, Kristina Shoonjans for the Hepa 1.6 cells, Pierre Gönczy for the U2OS cells, Sebastian Jessberger for KDEL-GFP, Anne-Laure Mahul for Mito-YFP, and Jean Grunberg for anti-LBPA.

## Author Contributions

**Conceptualization:** Patrick A. Sandoz, F. Gisou van der Goot, Mathieu Frechin.

**Data curation:** Patrick A. Sandoz, Mathieu Frechin.

**Formal analysis:** Patrick A. Sandoz, Christopher Tremblay, Mathieu Frechin.

**Funding acquisition:** F. Gisou van der Goot.

**Investigation:** Patrick A. Sandoz, Christopher Tremblay, Mathieu Frechin.

**Methodology:** Patrick A. Sandoz, Christopher Tremblay, Mathieu Frechin.

**Project administration:** Mathieu Frechin.

**Resources:** F. Gisou van der Goot, Mathieu Frechin.

**Software:** Mathieu Frechin.

**Supervision:** F. Gisou van der Goot, Mathieu Frechin.

**Validation:** Patrick A. Sandoz, F. Gisou van der Goot, Mathieu Frechin.

**Visualization:** Patrick A. Sandoz, Christopher Tremblay, Mathieu Frechin.

**Writing – original draft:** Mathieu Frechin.

**Writing – review & editing:** Patrick A. Sandoz, F. Gisou van der Goot, Mathieu Frechin.

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
