## [Editor Report · Decision Letter 0]

30 Sep 2019

Dear Mathieu, 

Thank you for submitting your revised manuscript entitled "Image-based analysis of living mammalian cells using label free 3D refractive index maps reveals new organelle dynamics and dry mass flux" for consideration as a Methods and Resources by PLOS Biology.

Your manuscript has now been evaluated by the PLOS Biology editorial staff as well as by the original academic editor and I am writing to let you know that we would like to send your submission out for external peer review.

Please re-submit your manuscript within two working days, i.e. by Oct 02 2019 11:59PM.

Kind regards,

Ines

--

Ines Alvarez-Garcia, PhD

Senior Editor

PLOS Biology

Carlyle House, Carlyle Road

Cambridge, CB4 3DN

+44 1223–442810

---

## [Decision Letter · Decision Letter 1]

21 Oct 2019

Dear Dr Frechin,

Thank you for submitting your revised Methods and Resources entitled "Image-based analysis of living mammalian cells using label free 3D refractive index maps reveals new organelle dynamics and dry mass flux" for publication in PLOS Biology. I have now obtained advice from the two original reviewers and have discussed their comments with the Academic Editor. 

Based on the reviews (attached below), we will probably accept this manuscript for publication, assuming that you will modify the manuscript to address the remaining concerns raised by the reviewers. Please address the remaining issues raised by Reviewer 2 by adding further details and clarifications to the text. In addition, the Academic Editor has sent us a marked-up vesion of the PDF highlighting a few gramatical errors in the text and a typo in Fig. 1 that should be amended (see file attached to this email or available in the editorial system).

We expect to receive your revised manuscript within two weeks. Your revisions should address the specific points made by each reviewer. In addition to the remaining revisions and before we will be able to formally accept your manuscript and consider it "in press", we also need to ensure that your article conforms to our guidelines. A member of our team will be in touch shortly with a set of requests. As we can't proceed until these requirements are met, your swift response will help prevent delays to publication.

Please note that you may have the opportunity to make the peer review history publicly available. The record will include editor decision letters (with reviews) and your responses to reviewer comments. If eligible, we will contact you to opt in or out.

Early Version

Sincerely,

Ines

--

Ines Alvarez-Garcia, PhD

Senior Editor

PLOS Biology

Carlyle House, Carlyle Road

Cambridge, CB4 3DN

+44 1223–442810

Reviewers’ comments

Rev. 1:

The revised manuscript is substantially improved. The authors have responded effectively to the points raised in the initial round of review, most notably by explaining the biological significance of the observed organelle rotation in mitotic cells. This is an excellent demonstration of the capabilities of the technique. The technique itself is described clearly and the discussion appropriately places the focus on the technique rather than the commercial implementation.

The present version of the manuscript is suitable for publication in PLOS Biology.

Rev. 2:

This is a revised manuscript presenting a label-free 3D refractive index imaging method based on holo-tomographic microscopy (HTM). I had to admit that the quality of the images and videos are very impressive. The HTM images have really good resolution and contrast, which compare favorably to bright-field, DIC and phase contrast microscopes. The authors tried to respond to the editor and reviewers’ comments and made a major revision to the manuscript. However, there are still several concerns with the current manuscript.

1. A similar manuscript from the authors was posted on BioRxiv in 2018. However, the authors listed are not exactly the same as the current submitted manuscript. Please explain.

https://www.biorxiv.org/content/biorxiv/early/2018/09/04/407239.full.pdf

2. Even with the additional details in the introduction, it is still not clear how the HTM system used in the current study was configured and how image reconstruction was performed. Figure 1a is not clear nor sufficient. The cited references are not up to date (e.g. the most recent reference in Ref. 1-10 are from 2013; many others were from 10-20 years ago).

3. If absolute values of the RI images were obtained, it would be important to show them in addition to the normalized images. A color bar can be added to the images to show the range of the absolute RI. It would be interesting to see the RI differences among various organelles, as well as variations between different cells.

4. No 3D views of the RI images were shown, although Figure 1b says so.

5. What is the depth range, axial resolution and temporal resolution of the HTM system? How small of a RI change can it detect?

6. The manuscript can be further improved with its readability. For example, the introduction can explain the difference and compare the pros and cons of HTM and other quantitative phase imaging methods. Technical details of the system can go into the method section instead of introduction. Connections between different experiments can be better explained. Any biological insights from these experiments would be appreciated.

7. There are also several mistakes/errors with uploaded figures and captions: Figures and captions for Extended Data Figure 2 and 3 are flipped; Caption of Extended Data Figure 7 refer to the Extended Data Figure 6. Caption for 8 points to Figure in 7; and caption for 6 points to Figure in 8; Extended Data Figure 5, what is shown in c?

---

## [Editor Report · Decision Letter 2]

15 Nov 2019

Dear Dr Frechin,

On behalf of my colleagues and the Academic Editor, Sandra L Schmid, I am pleased to inform you that we will be delighted to publish your Methods and Resources in PLOS Biology. 

Early Version

PRESS 

Kind regards,

Sofia Vickers

Senior Publications Assistant

PLOS Biology

On behalf of, 

Ines Alvarez-Garcia,

Senior Editor

PLOS Biology